# Coupling of saccade plans to endogenous attention during urgent choices

Allison T Goldstein, Terrence R Stanford, Emilio Salinas*

Department of Neurobiology and Anatomy, Wake Forest School of Medicine, Winston-Salem, United States

## eLife Assessment

This **important** study advances our understanding of the temporal dynamics and cortical mechanisms of eye movements and the cognitive process of attention. The evidence supporting the conclusions is **convincing** and based on measuring the time course of the eye movement-attention interaction in a novel, carefully-controlled experimental task. This study will be of broad interest to psychologists and neuroscientists interested in the dynamics of cognitive processes.

## Abstract

The neural mechanisms that willfully direct attention to specific locations in space are closely related to those for generating targeting eye movements (saccades). However, the degree to which the voluntary deployment of attention to a location necessarily activates a corresponding saccade plan remains unclear. One problem is that attention and saccades are both automatically driven by salient sensory events; another is that the underlying processes unfold within tens of milliseconds only. Here, we use an urgent task design to resolve the evolution of a visuomotor choice on a moment-by-moment basis while independently controlling the endogenous (goal-driven) and exogenous (salience-driven) contributions to performance. Human participants saw a peripheral cue and, depending on its color, either looked at it (prosaccade) or looked at a diametrically opposite, uninformative non-cue (antisaccade). By varying the luminance of the stimuli, the exogenous contributions could be cleanly dissociated from the endogenous process guiding the choice over time. According to the measured time courses, generating a correct antisaccade requires about 30 ms more processing time than generating a correct prosaccade based on the same perceptual signal. The results indicate that saccade plans elaborated during fixation are biased toward the location where attention is endogenously deployed, but the coupling is weak and can be willfully overridden very rapidly.

*For correspondence:
esalinas@wakehealth.edu

## Introduction

Primates make about four to five quick eye movements (saccades) every second. Before each movement, the oculomotor system selects a new target to look at depending on the match between the content of the current visual scene and the subject's internal state and current goals, i.e., what, if anything, the subject is looking for (*Itti and Koch, 2001*; *Theeuwes, 2010*; *Wolfe and Horowitz, 2017*). The rich dynamic between internal drives and what is out there in the world makes eye movements a good model system for understanding behavior at large.

Saccades are strongly coupled to visuospatial attention, which comprises a collection of mechanisms that regulate perception and thereby mediate the target selection process (*Carrasco, 2011*; *Maunsell, 2015*; *Moore and Zirnsak, 2017*). In general, the role of attention is to enhance stimuli that are potentially relevant and filter out those that are not. Neurons implicated in attention control

**eLife digest** You are attending a talk at a conference, eyes straight ahead and fixed on the speaker… yet you may in fact also be covertly monitoring your phone, hoping for a long-awaited message to flash on the screen. This ability to focus on something without directly looking at it is called spatial attention. It plays an essential role in everyday tasks, such as spotting keys on a cluttered desk or noticing when a traffic light changes.

Overlapping brain circuits control spatial attention and eye movements, creating tight links between the two processes. For example, shifting your gaze towards a specific location automatically leads you to pay at least partial attention to what unfolds at this spot. Whether the reverse is true, however, is less clear. In other words: when we are paying attention to something without looking at it, is our brain set to move our eyes towards this location?

To explore this question, Goldstein et al. designed a visual task that allowed them to track human participants' attention and eye movements moment by moment, and to unpick various factors affecting these processes. The volunteers fixed their gaze on the center of a screen, knowing that they also needed to pay attention to a certain location at the periphery where a cue was set to appear. The color of the cue determined whether the participants would then need to shift their gaze either towards or away from it – for example, they were instructed to look directly at a green cue but away from a magenta one.

These analyses showed that participants needed about 30 milliseconds less time to program an eye movement toward the cue – that is, to shift their gaze towards the location that they were already covertly monitoring. Such difference in processing time suggests that eye movements are biased towards the location on which attention is directed, but that this preference can still be overridden quickly.

By refining our understanding of the mechanisms underpinning attention, the findings by Goldstein et al. may help us better understand conditions like attention deficit hyperactivity disorder, where the brain struggles to engage and disengage with stimuli effectively.

and saccade generation are typically found within the same circuits. And importantly, the functional coupling between saccades and attention is bidirectional.

On one hand, the planning of a saccade automatically commits attentional resources to the saccade endpoint such that perceptual processing is enhanced at that location. This phenomenon, known as presaccadic attention, is firmly supported by both psychophysical (*Hoffman and Subramaniam, 1995*; *Kowler et al., 1995*; *Deubel and Schneider, 1996*; *Zhao et al., 2012*) and neurophysiological experiments (*Moore and Fallah, 2001*; *Moore and Fallah, 2004*; *Cavanaugh and Wurtz, 2004*; *Armstrong and Moore, 2007*; *Steinmetz and Moore, 2014*). Making a saccade implies attentional deployment.

The relationship in the opposite direction is less clear. With gaze held fixed, attention can be deployed willfully and covertly to a location in space, such that perceptual sensitivity is (typically) enhanced at that location (*Carrasco, 2011*; *Carrasco and Barbot, 2014*). This is referred to as endogenous attention, and there is evidence that its deployment impacts subsequent eye movements in two ways: it increases the probability that a saccade will be directed to the attended point (*Kustov and Robinson, 1996*; *Belopolsky and Theeuwes, 2009*; *Belopolsky and Theeuwes, 2012*), and if fixation is maintained, it alters the temporal and spatial characteristics of microsaccades (*Hafed and Clark, 2002*; *Engbert and Kliegl, 2003*). However, this link between endogenous attention and eye movements is not obligatory; the attentional locus and saccade endpoint can be dissociated (*Katnani and Gandhi, 2013*; *Steinmetz and Moore, 2014*; *Klapetek et al., 2016*), and microsaccades are only weak, unreliable markers of attentional allocation (*Yu et al., 2022*; *Willett and Mayo, 2023*). Furthermore, the effects of directing covert attention or a saccade to a given location can be quite distinct in terms of the activated neuronal populations (*Ignashchenkova et al., 2004*; *Thompson et al., 2005*; *Armstrong et al., 2009*), and may have slightly different perceptual consequences (for instance, on orientation or spatial frequency sensitivity; *Li et al., 2021*). Thus, the degree to which the early allocation of attention dictates subsequent saccade planning is still uncertain. This is the subject of the current study.

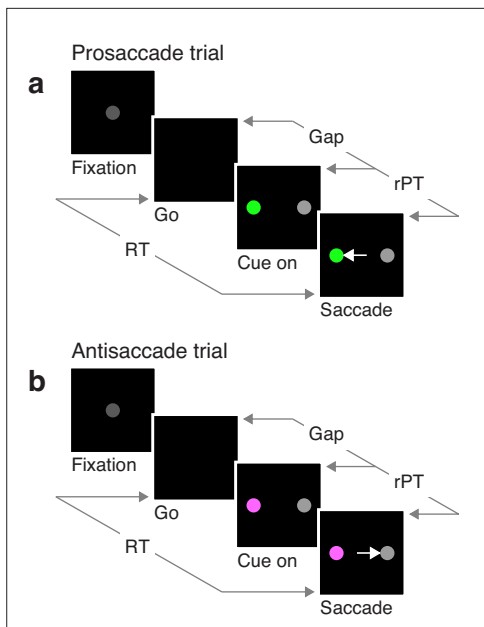

**Figure 1.** The endogenously driven pro/antisaccade task. (**a**) A prosaccade trial begins with a fixation period (Fixation, 500, 600, or 700 ms). The disappearance of the fixation point (Go) instructs the participant to look to the left or to the right within a reaction time (RT) window of 425–450 ms. After a variable time gap (Gap, 0–350 ms), a colored cue and a neutral non-cue appear simultaneously (Cue on) at locations that remain fixed for a block of trials. Thus, the participant always knows the cue location. The green color instructs the participant to look at the cue (Saccade). (**b**) An antisaccade trial proceeds in the same way as a prosaccade trial except that the magenta color instructs the participant to look at the non-cue. Pro- and antisaccade trials are randomly interleaved. In both, performance is dictated by the raw processing time (rPT), which is the amount of time during which the stimuli can be viewed and assessed before a response is initiated. The luminances of the stimuli can be set to balance the exogenous responses to the cue and non-cue (Experiments 1 and 2), or to create a bias toward either (Experiments 3 and 4).

There are three factors that make it difficult to mechanistically characterize how the willful allocation of spatial attention typically influences eye movements: (1) to reveal the effects of endogenous attention, most laboratory experiments require prolonged fixation, a condition that is somewhat artificial; (2) both attention and eye movements are drawn to salient events, so their underlying neural circuits both react automatically to some degree whenever a stimulus is presented or changed; and (3) the dynamic processes that underlie the allocation of attention and planning of saccades evolve very rapidly, with timescales on the order of a few tens of milliseconds. Thus, sustained fixation is regularly used to dissociate the endogenous deployment of attention from its overt counterpart (an eye movement) and from exogenous effects, which are transient. Normally, though, these three factors interact continuously, rapidly, and not necessarily in a fixed sequence.

To circumvent these hurdles, and be able to study how both forms of attention influence eye movements under less constrained conditions, we consider a task design in which the participant must make an eye movement either toward a peripheral cue (prosaccade) or in the opposite direction (antisaccade), toward an uninformative non-cue stimulus, depending on the color of the cue. This task has three main features. First, it is urgent. This means that the fixation requirement is brief and, because there is a reaction time (RT) deadline, motor plans must be initiated early, before the cue information is revealed. As detailed below, the resulting visuomotor dynamics are such that performance can be tracked with high precision as a function of time. Second, the luminances of the cue and non-cue are manipulated so that saccades are biased toward one stimulus, the other, or neither. This way, the contributon of exogenous attention can be precisely characterized and accounted for. And third, endogenous attention is deployed to a fixed (cue) location at the start of each trial. This may either shorten or lengthen the amount of time needed to make a successful choice, depending on the trial type (pro or anti) and on the coupling between endogenous attention and motor plans. Thus, the coupling strength can be inferred from the observed pattern of results.

For goal-directed movements, we report a consistent time delay of roughly 30 ms between prosaccade and antisaccade processing. This and other results indicate that, under relaxed fixation requirements, saccade plans are indeed coupled to the locus of endogenous attention – but can be voluntarily shifted quite rapidly.

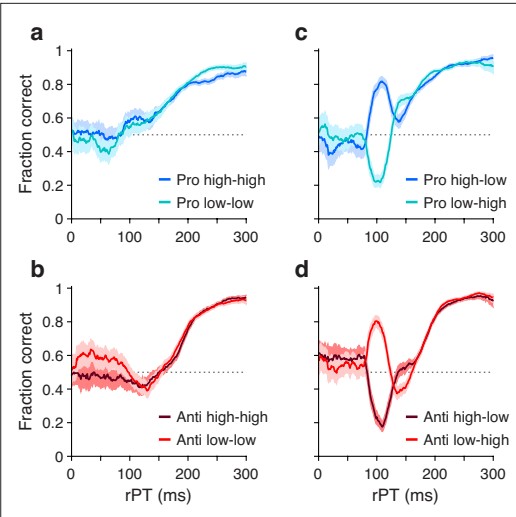

**Figure 2.** Performance in pro- (blue traces) and antisaccade trials (red traces) in four experiments where cue and non-cue luminance was varied. Each panel plots two tachometric curves. Each point on a curve indicates the fraction of correct choices for all the trials falling within a given processing time (rPT) bin (bin width=31 ms). (**a**) Performance in pro trials in Experiments 1 (high luminance cue, high luminance non-cue) and 2 (low luminance cue, low luminance non-cue). (**b**) As in a, but for anti trials. (**c**) Performance in pro trials in Experiments 3 (high luminance cue, low luminance non-cue) and 4 (low luminance cue, high luminance non-cue). (**d**) As in c, but for anti trials. Luminance combinations for cue and non-cue are indicated for each curve. In all panels, data are pooled across participants that met a performance criterion ($n = 11$; Methods). Error bands indicate 95% confidence intervals (CIs) across trials, from binomial statistics.

The online version of this article includes the following figure supplement(s) for figure 2:

**Figure supplement 1.** Performance in easy trials and inclusion criterion.

**Figure supplement 2.** Tachometric curves from unreliable performers.

**Figure supplement 3.** Prosaccade versus antisaccade performance aggregated across all the participants.

**Figure supplement 4.** Individual participants demonstrate varying degrees of bias and exogenous capture.

# Results

## A task design for revealing how endogenous attention modulates saccade plans

In earlier studies (*Salinas et al., 2019*; *Goldstein et al., 2022*), we used time pressure to characterize the contributions of exogenous and endogenous mechanisms to elementary visuomotor choices involving a single stimulus. By analyzing performance in an urgent prosaccade task (look toward a cue) and an urgent antisaccade task (look away from a cue), we found that exogenous and endogenous signals acted at different times and independently of each other. Specifically, the exogenous response to the cue onset manifested as 'captured' saccades, eye movements toward the cue that were, by all accounts, involuntary: they occurred early (~100 ms after cue onset), were strongly stereotyped across individuals, highly sensitive to luminance, and largely impervious to task rules or top-down control. In contrast, the endogenous response manifested as a sustained rise toward high-performance accuracy that was consistent with a deliberate process: the rise occured slightly later (~150 ms after cue onset), it was more variable across individuals, and was aligned with the task-defined goal.

In an effort to further isolate the endogenous component of spatial attention and characterize its impact on saccade planning, we developed a new task, the endogenously driven pro/antisaccade (EPA) task. This is, again, a two-alternative, urgent paradigm in which the participant must either look at a cue stimulus or look away from it, but there are two new elements. First, endogenous attention is deployed to a fixed location, so its influence on competing saccade plans (aligned or misaligned with that location) can be assessed. And second, the exogenous contribution is either unbiased or biased in favor of one or another choice, so that its effect can be cleanly identified. It is worth noting that in standard, non-urgent tasks, the cue is presented either before or simultaneously with the go signal, so the short-lived exogenous response is removed essentially by waiting for it to dissipate before the voluntary choice is made. However, our goal was to study the real-time interaction between attention and saccade planning over its natural timescale, which is on the order of a few tens of milliseconds, and the urgent nature of the paradigm is critical for this (*Stanford and Salinas, 2021*).

In the EPA task (*Figure 1*), the participant starts by fixating on a central spot (Fixation), and the offset of this spot is the go signal (Go) indicating that a saccade must be made to either the right or the left within 450 ms. At this point, the correct option has not been revealed yet, but if the saccade is to be made on time, the participant must attempt to respond nonetheless. After a variable,

unpredictable gap interval (Gap), both a cue and a non-cue appear simultaneously (Cue on). The participant is instructed to look at the cue if the cue is green (pro trials; *Figure 1a*), and to look at the non-cue if the cue is magenta (anti trials; *Figure 1b*). The cue color is selected randomly on each trial, but the cue and non-cue locations are fixed for each block of trials.

This task has three critical aspects. First, the urgency requirement (RT≤450 ms). Because of time pressure, the participants' responses range from guesses (uninformed saccades initiated before or soon after cue onset) at one extreme to fully informed choices (saccades initiated well after cue onset) at the other. The quantity that determines the degree to which a specific choice is informed, or the probability that it is informed, is the raw processing time (rPT), which is the time interval between the onset of the cue and the onset of the saccade (*Figure 1*, rPT). The rPT corresponds to the cue viewing time in each trial. Because this variable determines the probability of success, the main behavioral metric in the task is the 'tachometric curve', the curve that results when the fraction of correct choices is plotted as a function of rPT (*Figure 2*). As for other urgent tasks (*Salinas et al., 2019*; *Poth, 2021*; *Stanford and Salinas, 2021*; *Goldstein et al., 2022*; *Oor et al., 2023*), the tachometric curve depicts the evolution of the subject's choice on a moment-by-moment basis, and the details of this evolution can be uniquely revealing of the underlying attentional dynamics. For now, it suffices to note that, for all such data (*Figure 2a–d*), performance typically goes from mostly guesses (~50% correct) at short rPTs (≲75 ms) to mostly informed choices (~90% correct) at long rPTs (≳250 ms). Throughout the rest of the Results we will analyze in detail how this transition between chance and asymptotic performance occurs in the EPA task under different conditions.

The second important aspect of the EPA task design is that, at the beginning of each trial, endogenous attention is always directed toward the cue location. This is because the cue remains on the same side for an entire block of trials, switching only across blocks (150 trials per block), so the participant always knows where the color cue will appear. The non-cue never changes, so it is not informative. This way, to make an informed choice, the participant must attend to the cue, determine its color, and make a saccade according to the task rule (look to the green cue; look away from the magenta cue). Notably, the urgent nature of the task means that, by the time the cue is revealed, saccade plans must already be ongoing (*Stanford et al., 2010*; *Salinas et al., 2010*; *Costello et al., 2013*). This creates ideal conditions for studying how endogenous attention and motor plans interact to yield an overt saccadic choice.

Finally, the third task element that is critical is the non-cue. Although the non-cue stimulus is never informative of the correct choice, it serves a purpose: to control whether attention is exogenously drawn to one side, the other, or neither. That is, by adjusting the relative luminance of the cue and non-cue, it is possible to bias the choice in either direction or, alternatively, to balance the opposing exogenous influences so that the net bias is approximately zero. This study comprises four experiments, each one corresponding to a different luminance combination for the cue and non-cue stimuli (Methods; *Table 1*). In Experiment 1 the cue and non-cue were of equally high luminance, whereas in Experiment 2 they were of equally low luminance. In both cases the exogenous influences were meant to offset each other. In Experiment 3 a high luminance cue was paired with a low luminance non-cue, whereas in Experiment 4 the values were reversed, so the non-cue was brighter than the cue. In these cases the exogenous influence was meant to create a lateral bias favoring either the cue (Experiment 3) or the non-cue (Experiment 4) side.

**Table 1.** Stimulus parameters.

| Stimulus | RGB vector | Luminance (cd/m²) |
|---|---|---|
| High luminance green (cue) | [0 0.88 0] | 48 |
| Low luminance green (cue) | [0 0.1067 0] | 0.25 |
| High luminance magenta (cue) | [0.935 0.255 0.935] | 48 |
| Low luminance magenta (cue) | [0.1247 0.034 0.1247] | 0.25 |
| High luminance gray (non-cue) | [0.61 0.61 0.61] | 48 |
| Low luminance gray (non-cue) | [0.0813 0.0813 0.0813] | 0.25 |

In the following sections, we first examine the effect of exogenous attention on performance. Then, having understood the role of this element, we continue onto the main subject of the study, which is how the early deployment of endogenous attention during fixation influences the subsequent saccadic choice.

## Harnessing exogenous capture

Eighteen human participants (11 female, 7 male) were recruited and performed all four experiments (Methods). Thus, each participant generated data and a corresponding tachometric curve from two trial types (pro or anti) times four experiments for a total of eight experimental conditions. For some analyses, participants were divided into two groups, reliable performers ($n = 11$) and unreliable performers ($n = 7$). A participant was included in the reliable-performer group if he or she met a performance criterion in all eight conditions; otherwise they were included in the unreliable-performer group (Methods; *Figure 2—figure supplement 1*).

For each of the eight experimental conditions, an aggregate tachometric curve was generated by pooling the data from the reliable performers (*Figure 2*). In this initial comparison, the resulting eight tachometric curves are shown in pairs sorted by task, prosaccade (blue curves) or antisaccade (red curves). Each pair of curves thus represents the fraction of correct saccades made after a given amount of cue viewing time to each target stimulus, the cue (in pro trials) or the non-cue (in anti trials). As such, the effects of stimulus luminance can be easily visualized.

As intended, exogenous capture was minimized when the luminances of the cue and the non-cue were the same (*Figure 2a and b*). In this case, for both pro (panel a) and anti trials (panel b), the fraction correct hovers near 0.5 for processing times around 100 ms, which is when the exogenous effect would be expected to be strongest (*Salinas et al., 2019*; *Goldstein et al., 2022*; *Oor et al., 2023*). For each task, the subsequent rise toward asymptotic performance at longer rPTs is largely the same for stimuli of low (light-colored traces) and high luminance (dark-colored traces). This confirms that, for the most part, exogenous biases due to simultaneous onsets cancel out when they have similar strengths but point in opposite directions. The gradual, approximately monotonic rise in accuracy that results in this case is interpreted as the behavioral manifestation of the endogenously guided choice process.

Also as intended, exogenous capture was reinstated when the cue and the non-cue differed in luminance (*Figure 2c and d*), with the exogenous signal always biasing the saccades toward the high-luminance stimulus. In pro trials (*Figure 2c*), the salient cue (dark curve) produced a sharp, transient increase in the probability of making a correct prosaccade, whereas the salient non-cue (light curve) produced a sharp, transient decrease. In anti trials (*Figure 2d*), the effect was nearly identical but opposite in sign: the salient cue (dark curve) produced a sharp, transient decrease in the probability of making a correct antisaccade, whereas the salient non-cue (light curve) produced a sharp, transient increase. All of these capture effects were short-lived, peaking at rPT≈100 ms and mostly disappearing for rPT≳150 ms; thereafter, the endogenously driven rise toward asymptotic performance was, once again, basically the same across luminance conditions.

These results confirm that exogenous and endogenous influences are dissociable across processing time, and that their effects on visuomotor performance are largely independent of each other.

## Consistency of exogenous capture across participants and conditions

Although the early exogenous response is stimulus-driven, it remains uninformed because it does not indicate what the correct choice is; it is simply a bias toward high salience. To verify this characterization of exogenous attention in the EPA task, we performed subsequent analyses on the data from all individual participants in Experiments 3 and 4.

First, we defined an rPT window where the exogenous response typically occurred (Methods). For this, we replotted the aggregate tachometric curves from Experiments 3 (*Figure 3a*) and 4 (*Figure 3c*) but pairing the curves obtained from pro (blue traces) and anti trials (red traces) from the same experiment. Note that the early deviations from chance that characterize involuntary capture go in opposite directions but otherwise follow similar trajectories up to 25 ms or so past the point of strongest capture. This indicates that the exogenous signal is invariant to task instructions, in agreement with prior results (*Goldstein et al., 2022*). The common rPT interval bracketed by the crossover points of

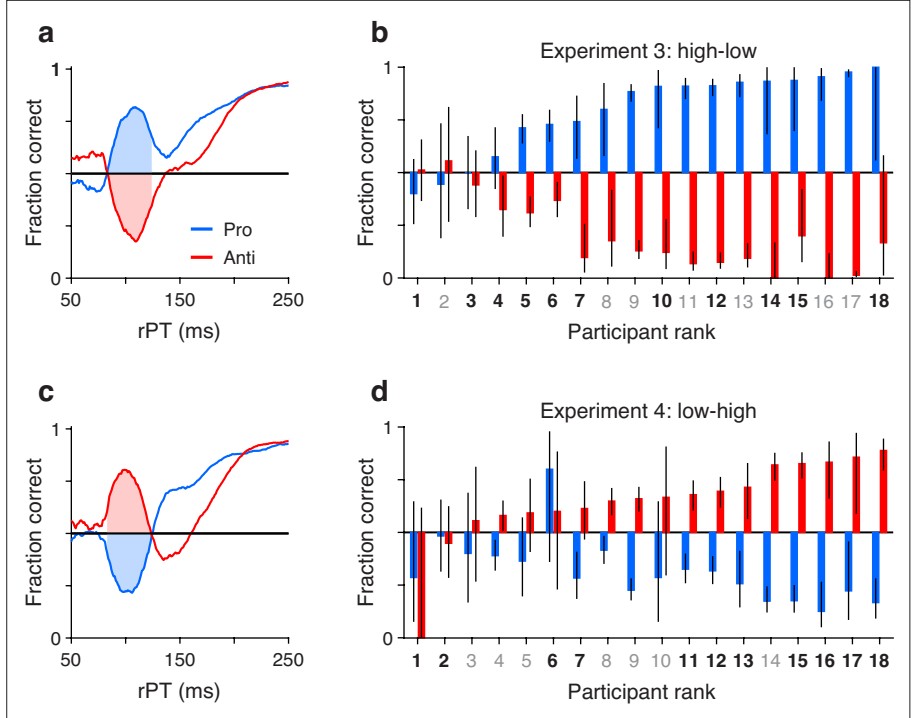

**Figure 3.** Exogenous capture across participants, tasks, and experiments. (**a**) Tachometric curves for pro (blue) and anti trials (red) in Experiment 3 (high luminance cue, low luminance non-cue). Data are from the reliable performers ($n = 11$). Shaded regions show the fixed raw processing time (rPT) window used for quantifying exogenous capture (83–124 ms). (**b**) Bar plots show fraction correct (y axis) in the exogenous capture window marked in a for all participants (x axis; $n = 18$). Pro- (blue) and antisaccade results (red) are from Experiment 3, with participants sorted by their fraction correct in pro trials. Color of axis labels indicates reliable (black) and unreliable (gray) participants. Black lines indicate 95% confidence intervals (CIs). (**c**) As in a, but for the data from Experiment 4 (low luminance cue, high luminance non-cue). (**d**) As in b, but for the data from Experiment 4. Participants are sorted by their fraction correct in anti trials. Note that capture is generally symmetric between pro and anti trials.

the mirrored trajectories was defined as the exogenous response window (rPT in 83–124 ms; *Figure 3a and c*, shaded areas).

Then, for each participant, we measured the fraction correct during the exogenous response window, sorting the trials separately for Experiments 3 (*Figure 3b*) and 4 (*Figure 3d*), and for pro (blue bars) and anti trials (red bars). With the resulting data arranged according to the magnitude of the effect, it is clear that the degree of exogenous capture varied quite dramatically across participants, from 0% to 100%. However, the magnitude of the capture as an absolute deviation from chance was statistically the same for pro and anti trials in both experiments. This was true when considering individual participants (for both Experiments 3 and 4, $p = 0.8$, $n = 18$, permutation test) or when pro and anti trials were pooled across participants (Experiment 3: $p = 0.5$, $n = 1564$ pro trials, $n = 1553$ anti trials, binomial test; Experiment 4: $p = 0.2$, $n = 1542$ pro trials, $n = 1489$ anti trials, binomial test). This result confirms that the early exogenous signal always tracks the higher luminance stimulus, and that the evoked exogenous response is the same regardless of the task rule.

## Possible linkage between endogenous attention and subsequent saccade planning

The above results circumscribe when and how exogenous attention biases saccadic choices in the EPA task. We now turn to the question of how the early deployment of endogenous attention might influence subsequent saccade planning, and how their interaction (or lack thereof) would manifest during task performance.

Consider two extreme possibilities. In the first scenario, saccade plans are entirely decoupled from endogenous attention. Thus, at the start of each trial, attention is endogenously deployed to the

cue location, but this does not bias the ensuing saccade plans. Right after the go signal, uninformed plans can be as easily initiated to the cue as to the non-cue; and later on, after the cue color has been resolved, they can be as easily redirected (by perceptual information) toward the cue or toward the non-cue. These decoupled dynamics give rise to two specific predictions. (1) On average, making an informed saccade toward the cue (in pro trials) should require the same amount of processing time as making an informed saccade toward the non-cue (in anti trials) – because once the cue color has been resolved, the resulting perceptual signal should be able to guide the developing motor selection process with equal effectiveness toward either stimulus. And (2), guesses toward the cue should, on average, be just as likely as guesses toward the non-cue. That is, internally generated saccade plans that advance rapidly before the cue color is resolved should not be systematically biased by endogenous attention being focused on the cue.

In the second scenario, saccade plans are strongly coupled to endogenous attention. Now, deploying endogenous attention implies a high degree of concomitant motor preparation. In this case, the initial uninformed plans generated right after the go signal are heavily biased toward the cue because attention is pointing there already; and the same thing is true later on: after the cue color has been resolved, plans are more easily redirected toward the cue than toward the non-cue. The specific predictions under strong-coupling dynamics are straightforward: (1) on average, making a perceptually informed saccade toward the cue (in pro trials) should require less processing time than making a perceptually informed saccade toward the non-cue (in anti trials), and (2) guesses should be predominantly directed toward the stimulus that is attended initially, i.e., the cue.

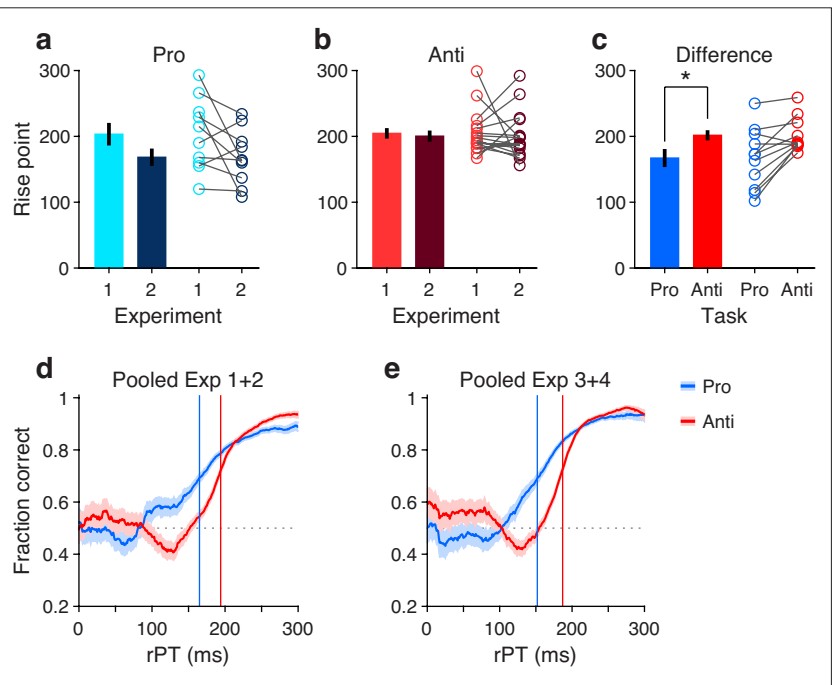

**Figure 4.** Antisaccades require more processing time than prosaccades. (**a**) Bars show mean rise point (±1 SE across participants) for pro trials in Experiments 1 and 2. Participants were included if their performance met a minimum modulation criterion in both experiments (Methods). Circles show data for individual qualifying participants ($n = 10$; $p = 0.08$ for the difference, from paired permutation test). (**b**) As in a, but for anti trials and corresponding qualifying participants ($n = 17$; $p = 0.80$). (**c**) Bars compare mean rise point (±1 SE across participants) in pro (raw processing time [rPT] = 167 ms) versus anti trials (rPT=202 ms). Data for each participant were pooled across Experiments 1 and 2. Participants were included if their combined performance met a minimum modulation criterion in both pro and anti trials ($n = 11$). Asterisk indicates $p = 0.008$ for the difference, from paired permutation test. (**d**) Tachometric curves for prosaccades (blue) and antisaccades (red). Data were pooled across Experiments 1 and 2 and across participants ($n = 11$; same group as in panel c). Blue and red lines denote rise points for pro (rPT = 165 ms) and anti trials (rPT = 194 ms). (**e**) As in d, but for data pooled across Experiments 3 and 4. Blue and red lines denote rise points for pro (rPT = 152 ms) and anti trials (rPT = 187 ms).

These extreme scenarios provide intuitive reference points for interpreting the behavioral data in the next sections, where the predictions are tested.

## The processing-time cost of an antisaccade

In summary, the intuition outlined in the previous section is that, if the initial deployment of endogenous attention necessarily entails some amount of motor preparation, then we would expect that, in the EPA task, generating an informed antisaccade should systematically require more processing time than generating an informed prosaccade. The question is how much. The difference could conceivably go from 0 ms (no coupling) to 100 ms (strong coupling), which is an estimate of the maximum amount of processing time needed to shift an ongoing motor plan from one location to a diametrically opposite one (*Salinas et al., 2019*; *Goldstein et al., 2022*).

To measure the rPT cost of an antisaccade relative to a prosaccade, we first considered the data from Experiments 1 and 2, in which the exogenous response was more muted. An important preliminary question was whether performance differed significantly between these two experiments. So, for each participant, we computed tachometric curves for pro and anti trials in Experiments 1 and 2, and fitted each curve with a sigmoid function (Methods). From each fitted curve, we determined the rise point, which is the rPT at which the curve reaches the midpoint between its minimum and maximum values; this quantity serves as a benchmark for when the saccadic choice becomes endogenously guided. Then we compared the rise points from Experiment 1 to those from Experiment 2 for pro trials (*Figure 4a*) and for anti trials (*Figure 4b*). Determining the rise point for an individual participant in any given experimental condition requires task performance to increase consistently as a function of rPT. Therefore, only participants that exceeded a modulation criterion were included in these comparisons, to ensure that the fits were acceptable and that the empirical curves were not flat (Methods; *Figure 4a–c*). Consistent with the pooled curves in *Figure 2a and b*, the rise points were similar across experiments for both prosaccades ($p = 0.08$, $n = 10$, permutation test) and antisaccades ($p = 0.8$, $n = 17$, permutation test). Given this, the data from Experiments 1 and 2 were combined.

Next, new tachometric curves for pro and anti trials were generated using the combined data, corresponding sigmoidal fits and rise points were obtained, and 11 participants were identified that met the modulation criterion in both task variants (a complementary analysis that includes all participants is discussed in a later section). Based on the data from these 11 participants, a comparison between rise points indicated that antisaccades consistently required more processing time than prosaccades to reach a comparable performance criterion (*Figure 4c*, circles). The sign of the effect was the same for 9 of the 11 qualifying participants ($p = 0.033$, binomial test). The average rise point was $167 \pm 13$ ms (mean $\pm$ 1 SE across participants) for prosaccades and $202 \pm 8$ ms for antisaccades, for a mean difference of 35 ms ($p = 0.008$, $n = 11$, permutation test; *Figure 4c*, bars). This is the average cost, in milliseconds of processing time, incurred for voluntarily programming a saccade away from the attended cue rather than toward it.

To further validate this difference in processing time, trials (again from Experiments 1 and 2) were pooled across the 11 qualifying participants to yield two aggregate tachometric curves, one for prosaccades (*Figure 4d*, blue trace) and another for antisaccades (*Figure 4d*, red trace). The resulting curves clearly show an earlier rise in prosaccade performance for informed choices (rPT $\gtrsim$ 150 ms), which have processing times that exceed the exogenous response window. Via sigmoidal fits, rise points were obtained for each of these curves (*Figure 4d*, vertical lines). Computed this way, the rise point from pro trials was 165 ms (in [158, 172] ms, 95% CI from bootstrap), whereas for anti trials it was 194 ms (in [191, 196] ms). The difference of 29 ms is comparable to the mean difference of 35 ms obtained from individual rise points.

Finally, we conducted the same analysis based on aggregate data from the qualifying participants, but this time pooling trials from Experiments 3 and 4 (*Figure 4e*). The idea was that, because the exogenous effects in pro and anti trials were always opposite and of similar magnitude (*Figure 3b and d*), on average they would cancel out, leaving only the net effect of endogenous guidance at each point in time (or, for short rPTs, any motor biases in the early uninformed choices). Indeed, this procedure yielded aggregate tachometric curves that were qualitatively similar to those obtained from Experiments 1 and 2 (*Figure 4e*). In this case, the rise point for pro trials was 152 ms (in [146, 157] ms, 95% CI), for anti trials it was 187 ms (in [185, 189] ms), and the difference was 35 ms.

These analyses were based on the rise points of the fitted sigmoidal functions, but results were very similar when using a fixed performance criterion of 70% correct to define the typical processing time required to make an informed choice (Methods). In that case, the differences in processing time between pro and anti trials were 24 ms for the aggregate curves from Experiments 1 and 2, and 34 ms for the aggregate curves from Experiments 3 and 4. Results were also similar when the data from each experiment were kept separate. Finally, when these analyses were repeated but including the data from all the participants, with no exclusions, the rise points in Experiment 1 could not be reliably determined due to strong motor biases (visible in *Figure 2—figure supplement 2a and b*), but in all other cases the rise in antisaccade performance lagged that in prosaccade performance in agreement with the above numbers (*Figure 2—figure supplement 3*). Most notably, the informed antisaccades were consistently delayed even in Experiment 4, in which motor plans were strongly biased toward the non-cue by the early exogenous signal (*Figure 3c*, *Figure 2—figure supplement 3d*).

These results indicate that the rPT cost of an endogenously guided antisaccade relative to an endogenously guided prosaccade is about 25–35 ms, and support the hypothesis that saccade plans are, to a degree, obligatorily coupled to endogenous attention.

## Guesses are predominantly biased toward the attended cue

As outlined earlier, another potential indicator of coupling between endogenous attention and subsequent saccade plans is the fraction of guesses that are directed toward the attended cue versus the

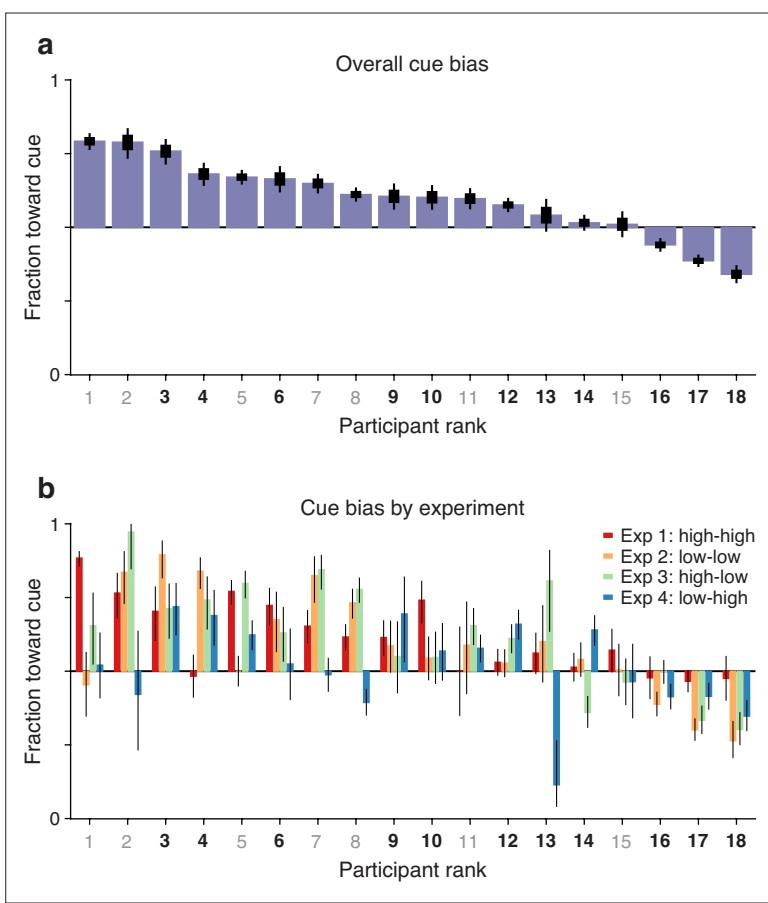

**Figure 5.** Guesses are biased toward the attended cue. Uninformed choices made at very short cue viewing times (raw processing time [rPT] ≤ 75 ms) are considered guesses. (**a**) Overall fraction of guesses made toward the cue (y axis), with participants (x axis) ranked by effect size. Results are for data aggregated across Experiments 1–4 and trial types (pro and anti). Color of axis labels indicates reliable (black) and unreliable (gray) participants. Thick and thin lines indicate 68% and 95% binomial confidence intervals (CIs). (**b**) Overall fraction of guesses made toward the cue in each experiment. Results are for data aggregated across trial types (pro and anti). Participant ranking is the same as above. Bar colors correspond to Experiments 1–4, as indicated. Lines correspond to 95% CIs.

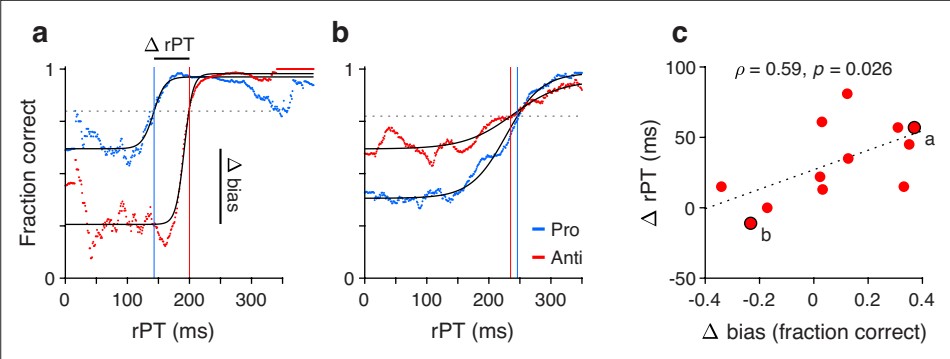

**Figure 6.** Covariation between motor bias and the processing-time cost of making an antisaccade. (**a**) Tachometric curves for pro (blue) and anti (red) trials from one participant whose motor bias was toward the cue. After fitting each curve with a sigmoid function (black traces), the halfway point between minimum and maximum values was determined. The highest halfway point (dashed line) served as a criterion. The mean time cost for an antisaccade, Δ rPT, was set as the difference (anti minus pro) between the raw processing times (rPTs) at which that criterion was reached (vertical lines). The corresponding motor bias magnitude for the participant, Δ bias, was set as the difference between the minimum values of the two fitted curves (pro minus anti). Each curve includes data from all experiments. (**b**) As in a, but for a participant whose motor bias was away from the cue. (**c**) Time cost of an antisaccade (y axis) as a function of motor bias magnitude (x axis). Each point corresponds to one participant ($n$ = 12 participants with acceptable fits). Data points from the two example participants in a and b are indicated. The dotted line corresponds to linear regression.

unattended non-cue. This is a straightforward measurement: for each participant, we considered all the trials (pro and anti) made at short rPTs (≤75 ms), before the cue and non-cue stimuli had had any effect on performance (see **Figures 2, 3a and c**), and calculated the fraction of choices made toward the cue. For each participant, this fraction indicates the preferred guessing side.

The results show that 15 of the 18 participants guessed predominantly toward the cue rather than toward the non-cue (**Figure 5a**). This motor bias was highly robust: the mean fraction of choices toward the cue was significantly above 0.5 when the averaging was across participants (**Figure 5a**; p = 0.00006, $n$ = 18, permutation test), across experimental conditions (**Figure 5b**; p = 0.00003, $n$ = 72, permutation test), or when the data were pooled across participants and experiments ($n$ = 9015 trials toward the cue, 7154 toward the non-cue, p = $10^{-48}$, binomial test). The data are as expected if saccade plans are partially coupled to the location where attention is endogenously deployed.

Having observed that guesses were predominantly directed toward the cue and that informed antisaccades generally required more processing time than informed prosaccades, we wondered whether the two effects were related or independent. To quantify this relationship, the processing time cost of an antisaccade was contrasted with the magnitude of the motor bias on an individual subject basis.

For a given participant, two tachometric curves were computed, one for pro and one for anti trials (**Figure 6a and b**). Because both the motor biases and the antisaccade costs were generally consistent across experiments, for this analysis the data were pooled across all four experiments. Each tachometric curve was fitted with a sigmoid function (**Figure 6a and b**, black curves; $n$ = 12 participants had both sigmoidal fits satisfying the minimum modulation criterion; see Methods). From each fitted curve, the halfway point between minimum and maximum values along the y axis was determined, and the highest of the two halfway values (from the pro or anti data) served as a criterion (**Figure 6a and b**, dashed lines). Finally, the mean processing-time cost of an antisaccade for the participant was taken as the difference between the rPT at which the antisaccade curve attained the criterion (**Figure 6a and b**, red vertical lines) minus the rPT at which the prosaccade curve attained it (**Figure 6a and b**, blue vertical lines). The motor bias of the participant was computed from the same sigmoidal fits; it was equal to the difference between the minimum value of the pro curve minus the minimum value of the anti curve.

Contrasting the processing-time cost of an antisaccade with the magnitude of the motor bias (**Figure 6c**) revealed a positive association of moderate strength between them (Pearson correlation, $\rho$ = 0.59, p = 0.026 from two-sided permutation test). This suggests that when the early attentional bias toward the cue is strong, it is more difficult for participants to produce an informed antisaccade

later on. Importantly, however, the initial motor bias may contribute to but does not explain the rPT cost of an antisaccade. As indicated by the offset of the regression line in *Figure 6c* (dotted line), the expected cost is 27 ms for zero bias. This agrees with data discussed earlier (*Figures 3a, c, 4d and e*, *Figure 2—figure supplement 3b and d*), which showed that there is a cost even for combinations of participants and experiments for which the bias is either minimal or slightly away from the cue. So, in general, producing an informed antisaccade required about 30 ms more of processing time than producing a similarly informed prosaccade, but this number varied by an additional amount in proportion to the strength of the initial bias toward or away from the cue.

## Higher efficiency for prosaccades versus antisaccades

The analyses above determined the processing-time cost of making an antisaccade away from the attended cue instead of a prosaccade toward it, everything else being equal. Here, we revisit this issue but from a different perspective. Instead of looking at the processing time needed to achieve a set performance criterion, we consider the reverse. Now we ask, given a fixed amount of processing time sufficient for motor plans to be partially informed, are prosaccades generally more successful than antisaccades? This alternate analysis does not exclude any participants. It is interesting not only

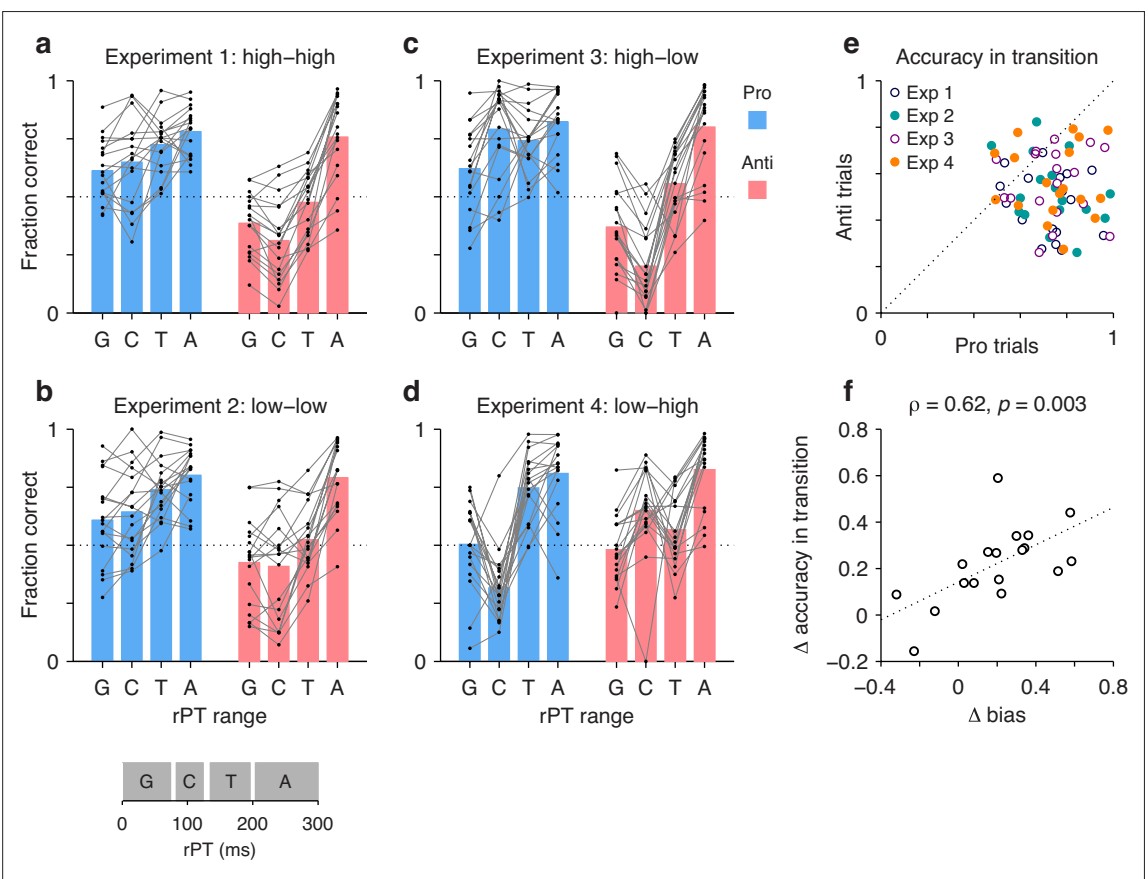

**Figure 7.** The transition from uninformed to informed performance is more rapid for prosaccades than for antisaccades. Processing times were divided into four non-overlapping ranges: a guessing range (G, raw processing time [rPT] ≤ 75 ms), a capture range (C, 83 ≤ rPT ≤ 124 ms), a transition range (T, 135 ≤ rPT < 200 ms), and an asymptotic range (A, rPT ≥ 200 ms); see inset at bottom. The fraction of correct choices was then computed separately for pro and anti trials in each experiment, in each rPT window, and for each participant. Responses in the T and A windows are informed by the cue color, whereas those in the G and C windows are not. (**a**) Results in Experiment 1. Fraction correct is shown for each of the four rPT windows. Black dots indicate data from individual participants ($n = 18$); blue and red bars show mean values for pro and anti trials, respectively, averaged across participants. The dotted line indicates chance performance. (**b–d**) As in a, but for Experiments 2–4. (**e**) Performance in anti trials (y axis) versus pro trials (x axis) with processing times in the transition range. Different symbols correspond to data from Experiments 1–4, as indicated, with one data point per participant. The dotted line indicates equality. Given the same amount of processing time, performance was typically higher during prosaccades than during antisaccades. (**f**) Differences between pro and antisaccade performance in the transition range (y axis) compared to those in the guessing range (x axis). Each point represents one participant ($n = 18$) with data pooled across experiments. The dotted line corresponds to linear regression.

as a verification of the above results, but also because it makes the variability in performance across individuals and experiments easier to appreciate.

In this case, individual trials were grouped according to rPT into four non-overlapping ranges (*Figure 7*, inset at bottom): a guessing range (rPT ≤ 75 ms), a capture range (83 ≤ rPT ≤ 124 ms), a transition range (135 ≤ rPT < 200 ms), and an asymptotic range (rPT ≥ 200 ms). Then, for each range, the fraction of correct choices was computed separately for pro and anti trials for each participant and each experiment. The results show how performance progresses over time in each condition (*Figure 7a–d*; black dots indicate individual participants). The procedure can be thought of as a simpler, discretized version of the tachometric curve for which only four time bins are considered, but the patterns discussed earlier are still recognizable. For instance, in Experiments 1–3, during pro trials guesses tend to be more successful than chance (*Figure 7a–c*, blue bars, G range), whereas during anti trials they tend to be less successful than chance (*Figure 7a–c*, red bars, G range). This, of course, reflects the internal bias toward the cue. The salience-driven capture of saccades is also easily recognizable. In Experiment 3, for most participants, the fraction correct in the capture range is well above chance in pro trials (*Figure 7c*, blue bars, C range) but well below chance in anti trials (*Figure 7c*, red bars, C range); and the effect in Experiment 4 is just the reverse (*Figure 7d*; compare blue vs. red bars in C range). From these data, it is also easy to see that most participants improve their performance with increasing rPT, as their accuracy tends to be highest in the asymptotic range (*Figure 7a–d*, A range). Notably, as seen with the full tachometric curves (*Figure 2—figure supplement 4*), in all the experiments there is considerable variability across participants, even in the asymptotic range. Such variance at long rPTs reflects a range of bias strengths as well as varying degrees of difficulty with the task (consistent with the easy trials; *Figure 2—figure supplement 1*).

The main question in this case is, what happens during the transition range? This rPT range covers the part of the tachometric curve during which accuracy rises consistently but has yet to reach its eventual asymptote. The responses produced during this time interval can be interepreted as saccades that are partly but not yet fully informed by the cue color; or alternatively, the data can be thought of as a mixture of informed and uninformed saccades. In either case, the question is whether at the given amount of processing time prosaccades have an advantage in performance over antisaccades.

The answer is yes. The fraction of correct choices during the transition range was consistently higher for pro than for anti trials (*Figure 7e*). This was true not only when considering the data from all four experiments together ($p < 10^{-5}$, $n = 72$, permutation test), but also when considering the data from each experiment separately (in all cases, $p \leq 0.003$, $n = 18$, permutation test). Again, the most notable result is for Experiment 4: in that case, pro trials demonstrate significantly higher accuracy than anti trials in the transition range (*Figure 7e*, orange points) in spite of the fact that, during the preceding capture interval, pro trials were at a huge disadvantage (*Figure 7d*; compare blue vs. red bars in C and T ranges). Thus, as perceptual information originating at the cue location starts to guide the ongoing target selection process, generating a movement toward the cue itself is easier than generating a movement away from it.

Finally, the data in this format can also be used to re-examine the degree to which the rPT cost of making an endogenously guided antisaccade covaries with the early motor bias. In this case, we contrasted the difference between pro and anti performance (fraction correct) in the transition range with the difference between pro and anti performance in the guessing range. The results (*Figure 7f*) confirmed the trend seen earlier based on the direct calculation of processing-time costs (*Figure 6c*), namely, there was a positive correlation (Pearson correlation, $\rho = 0.62$, $p = 0.003$ from two-sided permutation test). This was the case with the data pooled across experiments, but consistent, positive trends were observed for all four experiments individually. The results again suggest that when motor plans are strongly biased toward the cue, additional processing time is needed to overcome such bias and generate an informed (anti) saccade away from the cue – although, as remarked before, a cost is expected even with zero bias (see regression line in *Figure 7f*).

A last, notable point is that the motor bias was not a reliable predictor of overall task performance during informed choices. Across participants, the mean accuracy averaged over pro and anti trials combined had a weak, non-significant correlation with the bias in both the transition ($\rho = 0.09$, $p = 0.7$) and asymptotic ($\rho = 0.07$, $p = 0.8$) ranges. Thus, the bias did not incur an obvious disadvantage in terms of average success in the task. Its main consequence was simply to induce an asymmetry in timing and accuracy between pro- and antisaccades (*Figures 6c and 7f*).

## Discussion

The aim of the present study was to investigate how the voluntary deployment of spatial attention to an informative cue influences a subsequent eye movement either toward the cue or away from it. Because the interaction between attention and saccade planning is generally very fast, we focused on the difference in processing time required by these two conditions; and because stimulus presentation always implies a certain degree of exogenous influence, we also aimed to distinguish the respective contributions of exogenous and endogenous signals to the saccadic choice process. By imposing urgency we could generate a psychophysical curve describing the evolution of this choice process with high temporal resolution, and by manipulating the relative luminances of the cue and a non-cue we could characterize the strength and temporal extent of the exogenous signals associated with stimulus onsets. In this way, we were able to isolate the endogenously driven responses and resolve a time delay of approximately 30 ms between informed (pro) saccades toward the attended cue location and informed (anti) saccades toward an unattended, diametrically opposed location. This delay was highly consistent; it occurred even when the exogenous signal produced a bias in favor of the slower response. In addition, we also found that when participants made fast guesses, their saccades were significantly more likely to be toward the attended cue than toward the unattended non-cue.

### The observed coupling is consistent with prior studies

The findings indicate that when attention is voluntarily but covertly deployed, a subsequent saccade is generally biased toward the attended location. Thus, oculomotor planning is coupled to endogenous attention. The coupling is relatively weak, though, because motor plans could be willfully shifted very rapidly (within ~30 ms) to a different location, and because there was relatively high variability across conditions for any given participant (*Figure 5b*, *Figure 2—figure supplement 4*). The strength of the effect also varied considerably across individuals (*Figures 6c and 7f*). Importantly, as mentioned in the Introduction, this coupling pertains to only one side of the bidirectional relationship between saccade planning and spatial attention. The attentional effects that occur just before a saccade is executed have been amply characterized (*Hoffman and Subramaniam, 1995*; *Kowler et al., 1995*; *Deubel and Schneider, 1996*; *Moore and Fallah, 2001*; *Moore and Fallah, 2004*; *Cavanaugh and Wurtz, 2004*; *Armstrong and Moore, 2007*; *Zhao et al., 2012*; *Steinmetz and Moore, 2014*), and at this point the data are unequivocal: planning a saccade commits attentional resources to the intended saccade endpoint (or nearby, depending on how the target selection process progresses; *Wollenberg et al., 2018*). In contrast, here we examined the complementary relationship, i.e., how the early deployment of spatial attention biases the next saccade. Consistent with data showing that endogenous attention and saccade planning can be dissociated (*Ignashchenkova et al., 2004*; *Thompson et al., 2005*; *Armstrong et al., 2009*; *Steinmetz and Moore, 2014*; *Klapetek et al., 2016*; *Li et al., 2021*), our results are indicative of coupling that is comparatively weak in this direction.

In this regard, our results are comparable to those of *Belopolsky and Theeuwes, 2009*, who used a task in which participants made saccades to a target location that did or did not coincide with the location of an attended symbolic cue. They measured saccadic RTs that were much longer than in our case (roughly 500–800 ms), and yet their derived time costs for making an eye movement to an attended cue location versus to an unattended non-cue location were remarkably similar to ours, on the order of 20–40 ms. The authors interpreted their data in comparison to prior results by drawing a distinction between the maintenance of attention and the shifting of attention, because the time cost they observed was most robust when participants had to shift their attention to the cue location just before making a saccade to the target. This distinction is consistent with our view of urgency: when the fixation point disappears early on, motor plans can start developing even if the target has not been determined, in which case they will reflect any internal biases, including a bias toward the currently attended location. However, such bias typically becomes invisible under non-urgent conditions, which promote longer fixations. According to *Belopolsky and Theeuwes, 2009*, this is because the incipient oculomotor program associated with an attention shift can be suppressed shortly thereafter.

### Exogenous-endogenous interactions are rich and fast

A striking aspect of our data is the sharp distinction drawn between exogenous and endogenous attentional components, and the richness of their dynamic. Based on prior studies (*Salinas et al., 2019*; *Goldstein et al., 2022*; *Oor et al., 2023*), we expected the exogenous signal to be brief,

involuntary, and luminance-driven. And indeed, we were able to manipulate exogenous, involuntary capture as intended by varying the luminances of the stimuli, with highly consistent results across participants and conditions (*Figures 2 and 3*). Specifically, we were able to bias the participants' choices toward the cue or toward the non-cue, or to minimize the bias. The results support the notion that exogenous and endogenous influences on saccade programming are fundamentally independent and dissociable based on processing time (*Goldstein et al., 2022*). However, the results revealed an interesting wrinkle: the effect of the exogenous signal is not simply to advance the motor plan congruent with it and then fade away, in which case one would expect that responses would simply transition monotonically from captured saccades to informed choices with increasing rPT. Instead, when saccades are strongly biased (Experiments 3 and 4), the tachometric curves demonstrate the corresponding capture but then (briefly) go in the opposite direction. This can be seen in the pro curves of Experiment 3 (*Figure 3a*, *Figure 2—figure supplement 2c*, *Figure 2—figure supplement 3c*, dark blue traces) and in the anti curves of Experiment 4 (*Figure 3c*, *Figure 2—figure supplement 2d*, *Figure 2—figure supplement 3d*, bright red traces); in both cases there is a sharp increase in accuracy during the capture window (approximately 83–124 ms) that is then partially offset immediately afterward, before the final rise toward asymptotic performance. This downward rebound cannot be attributed to the endogenous perceptual signal because it goes in the opposite direction. It is as if the exogenous response was automatically followed by a motor reaction in the opposite direction. Perhaps the oculomotor circuitry is such that an exogenous signal can rapidly trigger a saccade, but if it does not, then the corresponding motor plan is rapidly suppressed regardless of anything else. This idea suggests a symmetry with the effect of endogenous attention discussed in the previous paragraph: it appears that in both cases, exogenous and endogenous, a shift of spatial attention activates a motor plan that is either executed within a few tens of milliseconds (to trigger a saccade) or is rapidly suppressed. Such a mechanism would be consistent with the intrinsic tendency of attentional signals to oscillate (*Landau and Fries, 2012*; *Hogendoorn, 2016*; *Fiebelkorn and Kastner, 2019*).

The dissociation of exogenous and endogenous influences over time also presents an interesting symmetry with respect to the effects of attention on saccade trajectories across space. Some of the earliest indications of a tight link between attention and eye movements came from experiments showing that saccades deviate away from a cue location that was previously attended but was never itself the saccade target (*Sheliga et al., 1994*; *Sheliga et al., 1995*). This led to the hypothesis that the mechanisms responsible for deploying spatial attention and those involved in programming saccades are essentially the same (i.e. the premotor theory of attention; *Rizzolatti et al., 1987*; *Belopolsky and Theeuwes, 2012*). However, later studies showed that attention effects on saccade trajectories could be either repulsive or attractive (*Van der Stigchel et al., 2006*), and specifically demonstrated that saccade trajectories could deviate either toward a distracter or away from it depending on the time at which the distracter was shown (*Theeuwes and Godijn, 2004*; *McSorley et al., 2006*; *Giuricich et al., 2023*). This result recapitulates the idea discussed above but in the spatial domain: the exogenous effect of the distracter is, initially, to produce a motor plan toward it (attraction), but later on this plan is suppressed and a motor bias away from the distracter (repulsion) is observed instead.

## Different tasks, different exogenous and endogenous signals

Our EPA task differs in many ways from the classic prosaccade/antisaccade paradigm (*Antoniades et al., 2013*), in which only the presence of the cue matters, not its features, and the instruction to look toward or look away is known to the subject at the start of each trial. Are such differences important?

There are many ways to set up an experiment where the subject either looks at a relevant cue or away from it. Conversely, it is also possible to design a task where the behavior is essentially identical to that in the classic antisaccade task without ever introducing the notion of looking away from something (*Oor et al., 2023*). We think that, more than the specific task instructions or the structure of the event sequence, the fundamental factors that determine behavior in all of these cases are the magnitudes of the resulting exogenous and endogenous signals, and whether they are aligned or misaligned. Under urgent conditions, consideration of these elements and their relevant timescales explains behavior in a wide variety of tasks (*Stanford and Salinas, 2021*). Furthermore, a recent study (*Zhu et al., 2024*) showed that the spatial signal encoded by neurons in monkey prefrontal cortex during the antisaccade task can be accurately predicted from their stimulus- and saccade-related responses during a simpler task that involves no spatial conflict whatsoever (a memory guided

saccade task). That is, these two independent response components explained the evolving attentional pointer observed during a typical antisaccade trial. This indicates that, at the circuit level, the dynamics of target selection are dictated by the relative strengths of the exogenous and endogenous activations and their congruency in space and time, however such activations are generated. Thus, we would expect this premise to also be valid under more natural viewing. In that case, visual transients would simply be less predictable and their corresponding exogenous influences potentially more variable.

## Attentional dynamics are better resolved under time pressure

A critical conclusion from our work is that temporal dependencies are much more accurately resolved when the fixation requirement is removed earlier and processing time is considered (instead of RT). There are many examples of this. Prior studies have shown that the effect of salience on saccadic choices is transient (*Donk and van Zoest, 2008*; *Siebold et al., 2011*; *Anderson et al., 2015*), that stimulus-driven and goal-driven control signals on visual selection are distinct and can be temporally dissociated (*van Zoest et al., 2004*), and that in the transition between them there is a moment of ambiguity wherein choices are dictated by neither (*van Heusden et al., 2022*). And, of course, various forms of capture have been demonstrated, all occurring early, before endogenous control takes over (*Theeuwes, 1992*; *Theeuwes, 1994*; *Failing et al., 2015*; *Aagten-Murphy and Bays, 2017*; *Nissens et al., 2017*). These findings often span differences in RT between 150 and 450 ms. The tachometric curves presented here and in our preceding reports (*Salinas et al., 2019*; *Goldstein et al., 2022*; *Oor et al., 2023*) recapitulate these phenomena but on a much faster timescale. The data in *Figure 2c and d* are emblematic: they reveal an early salience-driven response (characterized by captured saccades) and a late goal-driven rise toward asymptotic performance (>90% correct) separated by a transition period of ambiguous control, each distinct phase evolving within a few tens of milliseconds. What matters most is not the full saccade latency, but rather the shorter period of time during which the relevant stimuli (cues, distracters) can be viewed and processed, and that is the rPT. By initiating the motor plans early and using rPT as a time base, the fast transitions from uninformed to informed choices (*Stanford et al., 2010*; *Stanford and Salinas, 2021*) or from exogenous to endogenous control (*Salinas et al., 2019*; *Goldstein et al., 2022*; *Oor et al., 2023*; *Zhu et al., 2024*) are exposed with more clarity.

The broader lesson is that spatial attention dynamics in oculomotor circuits can vary more rapidly and abruptly than is generally appreciated. Saccades can go from being predominantly triggered in one direction to predominantly triggered in the opposite direction easily within 30–50 ms, and more than one such shift may occur in sequence (*Figure 2c and d*, *Figure 2—figure supplement 2c and d*, *Figure 2—figure supplement 4c*) as a consequence of distinct mechanisms interacting. The implications of these rich dynamics to more naturalistic visuomotor behaviors is an important avenue to explore in future studies.

## Methods

Methods were generally similar to those in preceding studies (*Salinas et al., 2019*; *Goldstein et al., 2022*). Here, we highlight key experimental procedures and details of the data analysis.

### Subjects

Experimental data were collected from 18 healthy human volunteers, 7 male and 11 female, with a median age of 28 years (range, 24–63). Subjects had normal or corrected-to-normal vision. All participants provided informed written consent before beginning the study. Participants were recruited from the Wake Forest University School of Medicine, Wake Forest University, and Atrium Health Wake Forest Baptist communities. All procedures were conducted with the approval of the Institutional Review Board of Wake Forest University School of Medicine (IRB00035241).

### Setup

The experiments were conducted in a semi-dark room. Participants sat in a height-adjustable chair with their chin and forehead supported on a desk-mounted head support. Stimuli were presented on a VIEWPixx LED monitor (VPixx Technologies Inc, Saint Bruno, Quebec, Canada; 1920 × 1200 screen

resolution, 120 Hz refresh rate, 12-bit color) at a distance of 57 cm. Eye position was measured and recorded using the EyeLink 1000 infrared camera and tracking system (SR Research, Ottawa, Canada; 1000 Hz sampling rate). Stimulus presentation and data acquisition were controlled using Matlab (The Mathworks, Natick, MA, USA) and the Psychtoolbox 3.0 package (*Brainard, 1997*; *Kleiner et al., 2007*).

## Behavioral tasks

The pro- and antisaccade tasks are similar to urgent tasks used in prior studies (*Salinas et al., 2019*; *Goldstein et al., 2022*; *Oor et al., 2023*), and require participants to make a perceptual judgment, a color discrimination in this case, while oculomotor plans are already ongoing. The sequence of events was the same in pro and anti trials, as described in *Figure 1*. A trial began with the onset of a gray central fixation point (RGB vector [0.25 0.25 0.25]) on a black screen. After fixation was maintained within a window (4° diameter) for a required time interval (500, 600, or 700 ms, randomly sampled), the fixation point was extinguished. The fixation point offset instructed the participant to make a saccade within 450 ms (or 425 ms for a few participants), and marked the start of the gap interval. Once the gap interval elapsed, the cue (green or magenta) and non-cue (gray) stimuli were shown, one on the left and the other on the right (at –8° and 8° along the horizontal). Participants were instructed to look at the cue whenever it was green (pro trial) and look at the non-cue whenever the cue was magenta (anti trial). Once a saccade was initiated, the cue and non-cue stimuli remained on the screen for 200 ms and then disappeared. A new trial began after an intertrial interval of 350 ms.

For all experiments, the cue location was constant throughout each block of 150 trials, and was thus known to the participant. Across blocks, it switched between left and right locations. In each trial, the color of the cue (green or magenta) was randomly sampled. The cue and non-cue were circles of 1.5° diameter. Gap durations were –200, –100, 0, 75, 100, 125, 150, 175, 200, 250, and 350 ms, and were randomly sampled across trials. So called 'easy' trials (gap < 0) are non-urgent trials in which the cue and non-cue stimuli were presented before the go signal, and were interleaved with urgent trials (gap ≥ 0) throughout each block. The RT was measured as the interval between fixation point offset and saccade onset. The rPT, or cue viewing time, was measured as the interval between stimulus onset and saccade onset, and was computed as rPT = RT – gap. An auditory feedback beep was provided at the end of a trial if the saccadic choice was made within the allowed 450 ms RT window. No sound played if the limit was exceeded. No feedback was provided about the correctness of the choice. The task proceeded in blocks of 150 trials with 2–3 min of rest between blocks.

To ensure that the task rule was understood, all participants completed short practice blocks of easy trials before beginning the experimental blocks and data collection. Experimental sessions lasted 1 hr, and each participant completed 8 blocks of trials in each of 8 sessions. Each participant completed 16 blocks per experiment: 8 blocks where the cue was always on the left interleaved with 8 blocks with the cue always on the right. For clarity, we refer to green-cue trials as pro trials and magenta-cue trials as anti trials; in practice, however, for participants P1–P10, a green cue instructed a prosaccade and a magenta cue instructed an antisaccade, whereas the colors were reversed for participants P11–P18. No effects of color assignment were found in any of the analyses. The cue and non-cue stimuli could each be of high or low luminance. Luminance values were chosen that triggered strong or weak exogenous capture when using single, lone targets, respectively (*Salinas et al., 2019*; *Goldstein et al., 2022*). In Experiment 1, the cue and non-cue were both high luminance. In Experiment 2, the cue and non-cue were both low luminance. In Experiment 3, the cue was high luminance and the non-cue was low luminance. In Experiment 4, the cue was low luminance and the non-cue was high luminance. Respective RGB vectors and luminance values are shown in *Table 1* for each stimulus. Luminance was determined by a spectrophotometer (i1 Pro 2 from X-Rite, Inc, Grand Rapids, MI, USA). Experiments were run in the same sequence, 1 through 4, for all participants.

## Data analysis

Analyses were carried out in the Matlab computing environment (The MathWorks, Natick, MA, USA; version 2013b or higher, with the Statistics Toolbox), as detailed in previous publications (*Salinas et al., 2019*; *Goldstein et al., 2022*).

Saccades were detected based on a velocity criterion (40°/s). Trials with fixation breaks (aborts) or blinks were excluded from analysis. Trials with saccades that were close to vertical (beyond ±60° of the

cue or non-cue; <1% of all trials) were also excluded. There was no explicit amplitude criterion; applying one (for instance, excluding any saccades with amplitude <2°) produced minimal changes to the data. Overall, saccade amplitudes were distributed unimodally with a median of 7.7° of eccentricity and a 95% confidence interval (CI) of [3.7°, 9.7°]; for reference, choice targets were located at ±8° horizontally. Most saccades were directed to the choice targets; 95% of them were within ±14.2° of the horizontal plane.

The RT was measured as the time between the go signal (fixation offset) and the onset of the saccade, which was taken as the first time point after the go signal for which the eye velocity surpassed the 40°/s threshold. Trials were scored as correct or incorrect based on the direction of the first saccade made after the go signal. Completed trials were included even if they exceeded the allotted RT limit.

Results are based on the analysis of urgent trials (gap ≥ 0) only. Easy, non-urgent trials (gap < 0), which yielded predominantly long rPTs (>200 ms), were excluded from analysis because, timing wise, they could potentially correspond to a slightly different regime (go signal after cue onset). In any case, their inclusion was of no appreciable consequence to the effects of interest. Note, however, that non-urgent trials were used to determine whether participants met a performance criterion (see below). No data were excluded based on participant performance or identity.

Tachometric curves describe how the fraction of correct choices evolves as a function of time after cue onset, or rPT. This processing time period was computed as rPT = RT – gap for each trial. For each tachometric curve, trials were grouped into rPT bins that shifted every 1 ms. For each bin, the numbers of correct and incorrect responses were tallied and the fraction of correct responses was calculated from them, with CIs determined by binomial statistics (68% CIs for *Figure 2—figure supplement 4*, 95% CIs for all other figures). The bin width was 31 ms when data were aggregated across multiple participants and 41 ms for tachometric curves for single participants.

The tachometric curves in Experiments 1 and 2 were fitted with a continuous analytical function, a monotonically increasing sigmoidal curve, to extract key characteristic metrics from the empirical curves. The sigmoid function was

$$s(x) = B + \frac{A - B}{1 + \exp(-\frac{x - C}{D})} \tag{1}$$

where $B$ is the baseline, $A$ is the asymptote, $C$ is the rise point, and $D$ determines the slope of the rise. For the current analyses, the most important parameter is the rise point, which corresponds to the rPT at which the fraction correct is halfway between the baseline and the asymptote. This quantity is indicative of when the perceptual discrimination is completed. As an alternative characteristic time point, we considered the rPT at criterion, which is the rPT at which the fraction correct first exceeds a fixed criterion $\theta$. Results (e.g. *Figure 4*) using the rise point and the rPT at criterion with $\theta = 0.7$ were very similar.

Although the parameter $B$ represents the fraction correct at short rPTs, for which participants guess and overall performance must be at chance, it was not constrained to 0.5, as in some of our prior experiments (*Salinas et al., 2019*; *Goldstein et al., 2022*). This is because, in this case, the participants knew where the cue would appear, so they could develop a preference for guessing toward or away from it. After sorting the pro and anti trials, such preference would manifest as substantial deviations from chance in the early parts of the corresponding tachometric curves. Thus, to accommodate these internal biases, the $B$ parameter was allowed to vary during the fitting process.

To find the optimal parameter values ($A$, $B$, $C$, and $D$) that best characterized a given tachometric curve, we minimized the mean absolute error between the empirical curve and the fitted curve using the fminsearch function in Matlab. Confidence intervals were obtained for these values by bootstrapping (*Davison and Hinkley, 2006*; *Hesterberg, 2015*). The bootstrapping process involved resampling the original trials with replacement, refitting the resampled tachometric curve, calculating the new parameters, and repeating the process 1000–10,000 times to generate distributions for all four parameters. With those distributions at hand, 95% CIs were determined using the 2.5 and 97.5 percentiles. When comparing the mean difference (averaged over qualifying participants) between rise points across two conditions (*Figure 4a–c*), significance was determined via a paired permutation test (*Siegel and Castellan, 1988*) with 100,000 iterations.

## Performance criterion

For some analyses (*Figure 2*, *Figure 2—figure supplement 2*), participants were sorted based on a criterion that indicated how well they performed the task without time pressure (*Figure 2—figure*

*supplement 1*). This was done by determining the fraction correct in easy trials only, i.e., in trials with gap <0 ms. The fraction correct was calculated separately for easy pro and easy anti trials in each experiment, for a total of eight performance measurements per participant. Then, to be included in the analysis, a participant had to exceed a fraction correct of 0.7 in all eight measurements. Eleven participants (deemed reliable) met this performance criterion and seven did not (unreliable).

### Modulation criterion
To be able to reliably determine a rise point value, as in *Figure 4*, a tachometric curve was required to be (1) not flat and (2) well fit by the sigmoid function. These conditions were quantified with the curve modulation

$$\Delta s = \max \left( s(x) \right) - \min \left( s(x) \right) \tag{2}$$

and the fit error

$$e = \left\langle |s(x) - f(x)| \right\rangle \tag{3}$$

where $s(x)$ is the fitted sigmoid, $f(x)$ is the empirical tachometric curve, and the angle brackets indicate an average for rPTs in the range 0–300 ms. To combine the two conditions into a single score, we computed the ratio $\Delta s/e$. This quantity is large for curves that are both strongly modulated and well fit by the sigmoid function. Participants with an error ratio above a threshold (2.5) were included in the analysis of rise points.

### Exogenous response window
In Experiments 3 and 4, we measured the degree of early exogenous capture by calculating the fraction of correct trials within a fixed rPT window. This exogenous response window was set to 83–124 ms based on the tachometric curves from the reliable performers (*Figure 3a and c*, shaded areas). It was defined as the common interval where the response to the high luminance stimulus was consistently above that to the low luminance stimulus, when taking into consideration the pro and anti tachometric curves in both experiments. Then, having defined this window, the fraction correct inside the window was computed separately for each participant and for pro and anti trials in each experiment (*Figure 3b and d*). The exact window limits were not critical to the results.

### Statistics
For comparisons where each data point represents one participant or one experimental condition from one participant, significance was determined based on permutation tests for paired data (*Siegel and Castellan, 1988*) or equivalent randomization tests for non-paired data. For comparisons involving binary data (correct vs. incorrect), confidence intervals and significance values were determined using binomial statistics (*Agresti and Coull, 1998*).

## Acknowledgements
We thank Denise Anderson for technical and logistical assistance. Research was supported by the NIH through grant R21MH120784 from the NIMH, grant R01EY025172 from the NEI, and training grant T32NS073553-01 from the NINDS.

## Additional information

### Competing interests
Emilio Salinas: Reviewing editor, *eLife*. The other authors declare that no competing interests exist.

## Funding

| Funder | Grant reference number | Author |
|---|---|---|
| National Institute of Mental Health | R21MH120784 | Terrence R Stanford<br>Emilio Salinas |
| National Eye Institute | R01EY025172 | Terrence R Stanford<br>Emilio Salinas |
| National Institute of Neurological Disorders and Stroke | T32NS073553-01 | Allison T Goldstein |

The funders had no role in study design, data collection and interpretation, or the decision to submit the work for publication.

## Author contributions

Allison T Goldstein, Conceptualization, Software, Formal analysis, Validation, Investigation, Visualization, Methodology, Writing - review and editing; Terrence R Stanford, Conceptualization, Resources, Supervision, Funding acquisition, Validation, Investigation, Methodology, Project administration, Writing - review and editing; Emilio Salinas, Conceptualization, Data curation, Software, Formal analysis, Supervision, Funding acquisition, Validation, Investigation, Visualization, Methodology, Writing - original draft, Writing - review and editing

## Author ORCIDs

Allison T Goldstein ⓘ https://orcid.org/0000-0002-5475-5965
Terrence R Stanford ⓘ https://orcid.org/0000-0003-0759-5599
Emilio Salinas ⓘ https://orcid.org/0000-0001-7411-5693

## Ethics

All participants provided informed written consent before beginning the study. All procedures were conducted with the approval of the Institutional Review Board of Wake Forest University School of Medicine (IRB00035241).

Reviewer #2 (Public review): https://doi.org/10.7554/eLife.97883.3.sa1
Reviewer #3 (Public review): https://doi.org/10.7554/eLife.97883.3.sa2
Author response https://doi.org/10.7554/eLife.97883.3.sa3

# Additional files

## Supplementary files

• MDAR checklist

## Data availability

The trial-by-trial behavioral data that support the findings of this study are publicly available from Zenodo at https://doi.org/10.5281/zenodo.10729511. Matlab scripts for reproducing analysis results and figures are included as part of the shared data package.

The following dataset was generated:

| Author(s) | Year | Dataset title | Dataset URL | Database and Identifier |
|---|---|---|---|---|
| Goldstein AT, Stanford TR, Salinas E | 2024 | Dataset: Coupling of saccade plans to endogenous attention during urgent choices (1.0.0) | https://doi.org/10.5281/zenodo.10729511 | Zenodo, 10.5281/zenodo.10729511 |

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
