## [Editor Report · eLife Assessment]

This **important** study advances our understanding of the temporal dynamics and cortical mechanisms of eye movements and the cognitive process of attention. The evidence supporting the conclusions is **convincing** and based on measuring the time course of the eye movement-attention interaction in a novel, carefully-controlled experimental task. This study will be of broad interest to psychologists and neuroscientists interested in the dynamics of cognitive processes.

---

## [Referee Report · Reviewer #2 (Public review)]

Goldstein et al. provide a thorough characterization of the interaction of attention and eye movement planning. These processes have been thought to be intertwined since at least the development of the Premotor Theory of Attention in 1987, and their relationship has been a continual source of debate and research for decades. Here, Goldstein et al. capitalize on their novel urgent saccade task to dissociate the effects of endogenous and exogenous attention on saccades towards and away from the cue. They find that attention and eye movements are, to some extent, linked to one another but that this link is transient and depends on the nature of the task. A primary strength of the work is that the researchers are able to carefully measure the time course of the interaction between attention and eye movements in various well-controlled experimental conditions. As a result, the behavioral interplay of two forms of attention (endogenous and exogenous) are illustrated at the level of tens of milliseconds as they interact with the planning and execution of saccades towards and away from the cued location. Overall, the results allow the authors to make meaningful claims about the time course of visual behavior, attention, and the potential neural mechanisms at a timescale relevant to everyday human behavior.

---

## [Referee Report · Reviewer #3 (Public review)]

The present study used an experimental procedure involving time-pressure for responding, in order to uncover how the control of saccades by exogenous and endogenous attention unfolds over time. The findings of the study indicate that saccade planning is influenced by the locus of endogenous attention, but that this influence was short-lasting and could be overcome quickly. Taken together, the present findings reveal new dynamics between endogenous attention and eye movement control and lead the way for studying them using experiments under time-pressure.

The results achieved by the present study advance our understanding of vision, eye movements, and their control by brain mechanisms for attention. In addition, they demonstrate how tasks involving time-pressure can be used to study the dynamics of cognitive processes. Therefore, the present study seems highly important not only for vision science, but also for psychology, (cognitive) neuroscience, and related research fields in general.

I think the authors' addressed all of the reviewers' points successfully and in detail, so that I don't have any further suggestions or comments.

---

## [Author Response]

The following is the authors’ response to the original reviews.

**Reviewer #1 (Public Review):**
The main research question could be defined more clearly. In the abstract and at some points throughout the manuscript, the authors indicate that the main purpose of the study was to assess whether the allocation of endogenous attention requires saccade planning [e.g., ll.3-5 or ll.247-248]. While the data show a coupling between endogenous attention and saccades, they do not point to a specific direction of this coupling (i.e., whether endogenous attention is necessary to successfully execute a saccade plan or whether a saccade plan necessarily accompanies endogenous attention).

Thanks for the suggestion. We have modified the text in the abstract and at various points in the text to make it more clear that the study investigates the relationship between attention and saccades in one particular direction, first attentional deployment and then saccade planning.

Some of the analyses were performed only on subgroups of the participants. The reporting of these subgroup analyses is transparent and data from all participants are reported in the supplementary figures. Still, these subgroup analyses may make the data appear more consistent, compared to when data is considered across all participants. For instance, the exogenous capture in Experiments 1 and 2 appears much weaker in Figure 2 (subgroup) than Figure S3 (all participants). Moreover, because different subgroups were used for different analyses, it is often difficult to follow and evaluate the results. For instance, the tachometric curves in Figure 2 (see also Figure 3 and 4) show no motor bias towards the cue (i.e., performance was at ~50% for rPTs <75 ms). I assume that the subsequent analyses of the motor bias were based on a very different subgroup. In fact, based on Figure S2, it seems that the motor bias was predominantly seen in the unreliable participants. Therefore, I often found the figures that were based on data across all participants (Figures 7 and S3) more informative to evaluate the overall pattern of results.

Indeed, our intent was to dissociate the effects on saccade bias and timing as clearly as possible, even if that meant having to parse the data into subgroups of participants for different analyses. We do think conceptually this is the better strategy, because the bias and timing effects were distinct and not strongly correlated with specific participants or task variants. For instance, the unreliable participants were somewhat more consistently biased in the same direction, but the reliable participants also showed substantial biases, so the difference in magnitude was relatively modest. This can be more easily appreciated now that the reliable and unreliable participants are indicated in Figures 3 and 5. The impact of the bias is also discussed further in the last paragraphs of the Results, which note that the bias was not a reliable predictor of overall success during informed choices.

**Reviewer #3 (Public Review):**
(1) In this experimental paradigm, participants must decide where to saccade based on the color of the cue in the visual periphery (they should have made a prosaccade toward a green cue and an antisaccade away from a magenta cue). Thus, irrespective of whether the cue signaled that a prosaccade or an antisaccade was to be made, the identity of the cue was always essential for the task (as the authors explain on p. 5, lines 129-138). Also, the location where the cue appeared was blocked, and thus known to the participants in advance, so that endogenous attention could be directed to the cue at the beginning of a trial (e.g., p. 5, lines 129-132). These aspects of the experimental paradigm differ from the classic prosaccade/antisaccade paradigm (e.g. Antoniades et al., 2013, Vision Research). In the classic paradigm, the identity of the cues does not have to be distinguished to solve the task, since there is only one stimulus that should be looked at (prosaccade) or away from (antisaccade), and whether a prosaccade or antisaccade was required is constant across a block of trials. Thus, in contrast to the present paradigm, in the classic paradigm, the participants do not know where the cue is about to appear, but they know whether to perform a prosaccade or an antisaccade based on the location of the cue.The present paradigm keeps the location of the cue constant in a block of trials by intention, because this ensures that endogenous attention is allocated to its location and is not overpowered by the exogenous capture of attention that would happen when a single stimulus appeared abruptly in the visual field. Thus, the reason for keeping the location of the cue constant seems convincing. However, I wondered what consequences the constant location would have for the task representations that persist across the task and govern how attention is allocated. In the classic paradigm, there is always a single stimulus that captures attention exogenously (as it appears abruptly). In a prosaccade block, participants can prioritize the visual transient caused by the stimulus, and follow it with a saccade to its coordinates. In an antisaccade block, following the transient with a saccade would always be wrong, so that participants could try to suppress the attention capture by the transient, and base their saccade on the coordinates of the opposite location. Thus, in prosaccade and antisaccade blocks, the task representations controlling how visual transients are processed to perform the task differ. In the present task, prosaccades and antisaccades cannot be distinguished by the visual transients. Thus, such a situation could favor endogenous attention and increase its influence on saccade planning, even though saccade planning under more naturalistic conditions would be dominated by visual transients. I suggest discussing how this (and vice versa the emphasis on visual transients in the classic paradigm) could affect the generality of the presented findings (e.g., how does this relate to the interpretation that saccade plans are obligatorily coupled to endogenous attention? See, Results, p. 10, lines 306-308, see also Deubel & Schneider, 1996, Vision Research).

Great discussion point. There are indeed many ways to set up an experiment where one must either look to a relevant cue or look away from it. Furthermore, it is also possible to arrange an experiment where the behavior is essentially identical to that in the classic antisaccade task without ever introducing the idea of looking away from something (Oor et al., 2023). More important than the specific task instructions or the structure of the event sequence, we think the fundamental factors that determine behavior in all of these cases are the magnitudes of the resulting exogenous and endogenous signals, and whether they are aligned or misaligned. Under urgent conditions, consideration of these elements and their relevant time scales explains behavior in a wide variety of tasks (see Salinas and Stanford, 2021). Furthermore, a recent study (Zhu et al., 2024) showed that the activation patterns of neurons in monkey prefrontal cortex during the antisaccade task can be accurately predicted from their stimulus- and saccade-related responses during a simpler task (a memory guided saccade task). This lends credence to the idea that, at the circuit level, the qualities that are critical for target selection and oculomotor performance are the relative strengths of the exogenous and endogenous signals, and their alignment in space and time. If we understand what those signals are, then it no longer matters how they were generated. The Discussion now includes a paragraph on this issue.

(2) Discussion (p. 16, lines 472-475): The authors suppose that "It is as if the exogenous response was automatically followed by a motor bias in the opposite direction. Perhaps the oculomotor circuitry is such that an exogenous signal can rapidly trigger a saccade, but if it does not, then the corresponding motor plan is rapidly suppressed regardless of anything else.". I think this interesting point should be discussed in more detail. Could it also be that instead of suppression, other currently active motor plans were enhanced? Would this involve attention? Some attention models assume that attention works by distributing available (neuronal) processing resources (e.g., Desimone & Duncan, 1995, Annual Review of Neuroscience; Bundesen, 1990, Psychological Review; Bundesen et al., 2005, Psychological Review) so that the information receiving the largest share of resources results in perception and is used for action, but this happens without the active suppression of information.

The rebound seen after the exogenously driven changes is certainly interesting, and we agree that it could involve not only the suppression of a specific motor plan but also enhancement of another (opposite) plan. However, we think that, given the lack of prior data with the requisite temporal precision, further elaboration of this point would just be too speculative in the context of the point that we are trying to make, which is simply that the underlying choice dynamics are more rapid and intricate than is generally appreciated.

(3) Methods, p. 19, lines 593-596: It is reported that saccades were scored based on their direction. I think more information should be provided to understand which eye movements entered the analysis. Was there a criterion for saccade amplitude? I think it would be very helpful to provide data on the distributions of saccade amplitudes or on their accuracy (e.g. average distance from target) or reliability (e.g. standard deviation of landing points). Also, it is reported that some data was excluded from the analysis, and I suggest reporting how much of the data was excluded. Was the exclusion of the data related to whether participants were "reliable" or "unreliable" performers?

The reported results are based on all saccades (detected according to a velocity threshold) that were produced after the go signal and in a predominantly horizontal direction (within ± 60° of the cue or non-cue), which were the vast majority (> 99%). Indeed, most saccades were directed to the choice targets, with 95% of them within ± 14.2° of the horizontal plane. The excluded (non-scored) trials were primarily fixation breaks plus a small fraction of trials with blinks, which compromised saccade determination. There was no explicit amplitude criterion; applying one (for instance, excluding any saccades with amplitude < 2°) produced minimal changes to the data. Overall, saccade amplitudes were distributed unimodally with a median of 7.7° and a 95% confidence interval of [3.7°, 9.7°], whereas the choice targets were located at ± 8° horizontally. This is now reported in the Methods.

As far as data exclusion, analyses were based on urgent trials (gap > 0); non-urgent (gap < 0) trials were excluded from calculation of the tachometric curves simply because they might correspond to a slightly different regime (go signal *after* cue onset) and to long processing times in the asymptotic range (rPT in 200–300 ms) or beyond, which are not as informative. However, including them made no appreciable difference to the results. No data were excluded based on participant performance or identity; all psychometric analyses were carried out after the selection of trials based on the scoring criteria described above. This is now stated in the Methods.

(4) Results, p. 9, lines 262-266: Some data analyses are performed on a subset of participants that met certain performance criteria. The reasons for this data selection seem convincing (e.g. to ensure empirical curves were not flat, line 264). Nevertheless, I suggest to explain and justify this step in more detail. In addition, if not all participants achieved an acceptable performance and data quality, this could also speak to the experimental task and its difficulty. Thus, I suggest discussing the potential implications of this, in particular, how this could affect the studied mechanisms, and whether it could limit the presented findings to a special group within the studied population.

The ideal (i.e., best) analysis for determining the cost of an antisaccade for each individual participant (Fig. 4c) was based on curve fitting and required task performance to rise consistently above chance at long rPTs in both pro and anti trials. This is why the mentioned conditions on the fits were imposed. This is now explained in the text. This ideal analysis was not viable for all tachometric curves not necessarily because of task difficulty but also because of high variability or high bias in a particular experiment/condition. It is true that the task was somewhat difficult, but this manifested in various ways across the dataset, so attempting to draw a clean-cut classification of participants based on “difficulty” may not be easy or all that informative (as can be gleaned from Fig. S1). There simply was a range of success levels, as one might expect from any task that requires some nontrivial cognitive processing. Also note that no participants were excluded flat out from analysis. Thus, at the mentioned point in the text, we simply note that a complementary analysis is presented later that includes all participants and all conditions and provides a highly consistent result (namely, Fig. 7e). Then, in the last section of the Results, where Fig. 7 is presented, we point out that there is considerable variance in performance at long rPTs, and that it relates to both the bias and the difficulty of the task across participants.

**Reviewer #1 (Recommendations For The Authors):**
(1) I have some questions related to the initial motor bias:a) Based on Figure S3, which shows the tachometric curves using data from all participants, there only seems to be a systematic motor bias in Experiments 1 and 3 but no bias in Experiments 2 and 4. It is unclear to me why this is different from the data shown in Figure 7.

For the bars in Fig. 7, accuracy (% correct) was computed for each participant and then averaged across participants, whereas for the data in Fig. S3, trials were first pooled across participants and then accuracy was computed for each rPT bin. The different averaging methods produce slightly different results because some participants had more trials in the guessing range than others, and different biases.

b) Based on Figure 7 (and Figure S3), there was no motor bias in Experiment 4. Based on the correlations between motor bias and time difference between pro and antisaccades, I would expect that the rise points between pro and antisaccades would be more similar in this Experiment. Was this the case?

No. Figs. 3c and S3d show that the rise times of pro and anti trials for Experiment 4 still differ by about 30 ms (around the 75% correct mark), and the rest of the panels in those figures show that the difference is similar for all experiments. What happens is that Figs. 7 and S3 show that on average the bias is zero for Experiment 4, but that does not mean that the average difference in rise times is zero because there is an offset in the data (correlation is not the same as regression). The most relevant evidence is in Fig. 6c, which shows that, for an overall bias of zero, one would still expect a positive difference in rise times of about 25–30 ms. This figure now includes a regression line, and the corresponding text now explains the relationship between bias and rise times more clearly. Thanks for asking; this is an important point that was not sufficiently elaborated before.

c) If I understand correctly, the initial motor bias was predominantly observed in participants who were classified as 'unreliable performers' (comparing Figure S2 and Figure 2). Was there a correlation between the motor bias and overall success in the task? In other words: Was a strong motor bias generally disadvantageous?

Good question. Participants classified as ‘unreliable’ were somewhat more consistently biased in the same direction than those classified as ‘reliable’, but the distinction in magnitude was not large. This can be better appreciated now in Fig. 5 by noting the mix of black (reliable) and gray labels (unreliable) along the x axes. The unreliable participants were also, by definition, less accurate in their asymptotic performance in at least one experiment (Fig. S1). In general, however, this classification was used simply to distinguish more clearly the two main effects in the data (timing cost and bias). In fact, the motor bias was not a reliable predictor of performance during informed choices: across all participants, the mean accuracy in the asymptotic range (rPT > 200 ms) had a weak, non-significant correlation with the bias (ρ = ‒0.07, *p* = 0.7). So, no, the motor bias did not incur an obvious disadvantage in terms of overall success in the task. Its more relevant effect was the *asymmetry* in performance that it promoted between pro- and antisaccade trials (Fig. 6c). This is now explained at the end of the Results.

(2) One of the key analyses of the current study is the comparison of the rPT required to make informed pro and antisaccades (ll.246 ff). I think it would be informative for readers to see the results of this analysis separately for all four experiments. For instance, based on Figure 4a and b, it looks like the rise points were actually very similar between pro and antisaccades in Experiment 1.

We agree that the ideal analysis would be to compute the performance rise point for pro- and antisaccade curves for each experiment and each participant, but as is now noted in the text, this requires a steady and substantial rise in the tachometric curve, which is not always obtained at such a fine-grained level; the underlying variability can be glimpsed from the individual points in Fig. 7a, b. Indeed, in Fig. 4a, b the mean difference between pro and anti rise points appears small for Experiment 1 — but note that the two panels include data from only partially overlapping sets of participants; the figure legend now makes this more clear. Again, this is because the required fitting procedure was not always reliable in both conditions (pro and anti) for a given subject in a given experiment. Thus, panels a and b cannot be directly compared. The key results are those in Fig. 4c, which compare the rise points in the two conditions for the same participants (11 of them, for which both rise points could be reliably determined). In that case the mean difference is evident, and the individual effect consistent for 9 of the 11 participants (as now noted).

A similar comparison for Experiments 1 or 2 individually would include fewer data points and lose statistical power. However, on average, the results for Experiments 1 and 2 (separately) were indeed very similar; in both cases, the comparison between pro and anti curves pooled across the same qualifying participants as in Fig. 4c produced results that were nearly identical to those of Fig. 4d (as can be inferred from Fig. 2a, b). Furthermore, results for the four individual experiments pooled across all participants are presented in Figure S3, which shows delayed rises in antisaccade performance consistent with the single participant data (Fig. 4c).

(3) Figure 3: It would be helpful to indicate the reliable performers that were used for Figure 3a in the bar plots in Figure 3b. Same for Figures 3c and d.

Done. Thanks for the suggestion.

(4) Introduction: The literature on the link between covert attention and directional biases in microsaccades seems relevant in the context of the current study (e.g., Hafed et al., 2002, Vision Res; Engbert & Kliegl, 2003, Vision Res; Willett & Mayo, 2023, Proc Natl Acad Sci USA).

Yes, thanks for the suggestion. The introduction now mentions the link between attentional allocation and microsaccade production.

(5) ll.395ff & Figure 7f: Please clarify whether data were pooled across all four experiments for this analysis.

Yes, the data were pooled, but a positive trend was observed for each of the four experiments individually. This is now stated.

(6) ll.432-433: There is evidence that the attentional locus and the actual saccade endpoint can also be dissociated (e.g., Wollenberg et al., 2018, PLoS Biol; Hanning et al., 2019, Proc Natl Acad Sci USA).

True. We have rephrased accordingly. Thanks for the correction.

(7) ll.438-440: This sentence is difficult to parse.

Fixed.

**Reviewer #2 (Recommendations For The Authors):**
The manuscript is well-written and compelling. The biggest issue for me was keeping track of the specifics of the individual experiments. I think some small efforts to reinforce those details along the way would help the reader. For example, in the Figure 3 figure legend, I found the parenthetical phrase "high luminence cue, low luminence non-cue" immensely helpful. It would be helpful and trivial to add the corresponding phrase after "Experiment 4" in the same legend.

Thanks for the suggestion. Legends and/or labels have been expanded accordingly in this and other figures.

Line 314: "..had any effect on performance,..." Should there be a callout to Figure 2 here?

Done.

It wasn't clear to me why the specific high and low luminance values (48 and 0.25) were chosen. I assume there was at least some quick perceptual assessment. If that's the case or if the values were taken from prior work, please include that information.

Done.

**Reviewer #3 (Recommendations For The Authors):**
Minor points. Please note that the comments made in the public review above are not repeated here.(1) Introduction, p. 2, lines 41-45: It is mentioned that the effects of covert attention or a saccade can be quite distinct. I suggest specifying in what way.

Done.

(2) Introduction, p. 2, lines 46-47: It is said that the relation between attention and saccade planning was still uncertain and then it is stressed that this was the case for more natural viewing conditions. However, the discussed literature and the experimental approach of the current study still rely on experimental paradigms that are far from natural viewing conditions. Thus, I suggest either discussing the link between these paradigms and natural viewing in more detail or leaving out the reference to natural viewing at this point (I think the latter suggestion would fit the present paper best).

We followed the latter suggestion.

(3) Introduction (e.g. p. 3, lines 55-58): The authors discuss the effects that sustaining fixation might have on attention and eye movements. Recently, it has been found that maintaining fixation can ameliorate cognitive conflicts that involve spatial attention (Krause & Poth, 2023, iScience). It seems interesting to include this finding in the discussion, because it supports the authors' view that it is necessary to study fixation and eye movements rather than eye movements alone to uncover their interplay with attention and decision-making.

Thanks for the reference. The reported finding is certainly interesting, but we find it somewhat tangential to the specific point we make about strong fixation constraints — which is that they suppress internally driven motor activity, including biases, that are highly informative of the relationship between attention and saccade planning (lines 466‒472, 541‒561). Whether fixation state has other subtle consequences for cognitive control is an intriguing, important issue, for sure. But we would rather maintain the readers’ focus on the reasons why less restrictive fixation requirements are relevant for understanding the deployment of attention.

(4) Results, p. 9, lines 264-266: It is reported that "The rise points were statistically the same across experiments for both prosaccades (p=0.08, n=10, permutation test)...", but the p-value seems quite close to significance. I suggest mentioning this and phrasing the sentence a bit more carefully.

We now refer to the rise points as “similar”.

(5) Figure 7 a-d: It might help readers who first skim through the figures before reading the text to use other labels for the bins on the x-axis that spell out the name of the phase in the trial. It might also help to visualize the bins on the plot of a tachymetric function (in this case, changing the labels could be unnecessary).

Thanks for the suggestion. We added an insert to the figure to indicate the correspondence between labels and time bins more intuitively.

(6) Methods, p. 18, lines 566-567: On some trials, participants received an auditory beep as a feedback stimulus. As this could induce a burst of arousal, I wondered how it affected the subsequent trials.

This is an interesting issue to ponder. We agree that, in principle, the beep could have an impact on arousal. However, what exactly would be predicted as a consequence? The absence of a beep is meant to increase the urgency of the participant, so some effect of the beep event on RT would be expected anyway as per task instructions. Thus, it is unclear whether an arousal contribution could be isolated from other confounds. That said, three observations suggest that, at most, an independent arousal effect would be very small. First, we have performed multisensory experiments (unpublished) with auditory and visual stimuli, and have found that it is difficult to obtain a measurable effect of sound on an urgent visual choice task unless the experimental conditions are particularly conducive; namely, when the visual stimuli are dim and the sound is loud and lateralized. None of these conditions applies to the standard feedback beep. Second, because most trials are on time, the meaningful feedback signal is conveyed by the *absence* of the beep. But this signal to alter behavior (i.e., respond sooner) has zero intensity and is therefore unlikely to trigger a strong exogenous, automatic response. Finally, in our data, we can parse the trials that followed a beep (the majority) from those that did not (a minority). In doing so, we found no differences with respect to perceptual performance; only minor differences in RT that were identical for pro- and antisaccade trials. All this suggests to us that it is very unlikely that the feedback alters arousal significantly on specific trials, somehow impacting the tachometric curve (a contribution to general arousal across blocks or sessions is possible, of course, but would be of little consequence to the aims of the study).

(7) Methods, p. 18, lines 574-577: I suggest referring to the colors or the conditions in the text as it was done in the experiments, just to prevent readers being confused before reading the methods.

We appreciate the thought, but think that the study is easier to understand by pretending, initially, that the color assignments were fixed. This is a harmless simplification. Mentioning the actual color assignments early on would be potentially more confusing and make the description of the task longer and more contrived.

(8) Methods, p. 18, Table 1: Given that the authors had a spectrophotometer, I suggest providing (approximate) measurements for the stimulus colors in addition to the luminance (i.e. not just RGB values).

Unfortunately, we have since switched the monitor in our setup, so we don’t have the exact color measurements for the stimuli used at the time. We will keep the suggestion in mind for future studies though.

References

Oor EE, Stanford TR, Salinas E (2023) Stimulus salience conflicts and colludes with endogenous goals during urgent choices. iScience 26:106253.

Salinas E, Stanford TR (2021) Under time pressure, the exogenous modulation of saccade plans is ubiquitous, intricate, and lawful. Curr Opin Neurobiol 70:154-162.

Zhu J, Zhou XM, Constantinidis C, Salinas E, Stanford TR (2024) Parallel signatures of cognitive maturation in primate antisaccade performance and prefrontal activity. iScience. doi: https://doi.org/10.1016/j.isci.2024.110488.